# DIET-CP: Lightweight and Data Efficient Self Supervised Continued Pretraining

**Bryan Rodas**[*]
Fordham University
Quantitative Finance Department
brodas1@fordham.edu

**Natalie Montesino**[*]
Rutgers University
Electrical and Computer Engineering Department
nlm128@scarletmail.rutgers.edu

**Jakob Ambsdorf**[*]
Pioneer Centre for AI
University of Copenhagen
jaam@di.ku.dk

**David Klindt**
Cold Spring Harbor Laboratory
klindt@cshl.edu

**Randall Balestriero**
Brown University
Computer Science Department
rbalestr@brown.edu

## Abstract

Continued pretraining offers a promising solution for adapting foundation models to a new target domain. However, in specialized domains, available datasets are often very small, limiting the applicability of SSL methods developed for large-scale pretraining, and making hyperparameter search infeasible. In addition, pretrained models are usually released as backbone-weights only, lacking important information to continue pretraining. We propose to bridge this gap with DIET-CP, a simple continued pretraining strategy, where any strong foundation model can be steered towards the new data distribution of interest. DIET-CP relies on a very simple objective, requires no labels, and introduces no more hyperparameters than supervised finetuning. It is stable across data modalities and backbone choices, while providing a significant performance boost for state-of-the-art models such as DINOv3 using only 1000 images.

## 1 Introduction

Foundation models promise robust features for a variety of tasks and domains, powered by increasingly larger and diverse pretraining datasets. However, despite the all-time-high transfer-learning performance of pretrained models, there still remains a margin to expert models trained within one domain and modality [1, 2]. Continued pretraining on the target domain is a potential solution to this problem [3, 4, 5]. However, while state-of-the-art foundation models such as DINOv3 [6] can–in theory–be further pretrained, researchers and practitioners are often facing three problems that make this approach infeasible: (1.) Models are released as backbone weights only, missing crucial information to continue pretraining, such as teacher weights or optimizer state. [6, 7] (2.) State-of-the-art self-supervised learning methods introduce a multitude of hyperparameters, which are costly and difficult to tune for the target domain, or even intractable if only few samples are available. [8] (3.) The pretraining methods themselves are optimized for large-scale datasets, while target datasets are significantly smaller [9].

Motivated to overcome these practical hurdles, we propose DIET-CP: A label-free and efficient method for steering foundation models towards a new distribution of interest. Our method relies on a very simple objective that requires only the pretrained backbone, that is free of additional hyperparameters, stable over data modality and backbone employed, all while providing a significant

---

[*]Equal contribution.

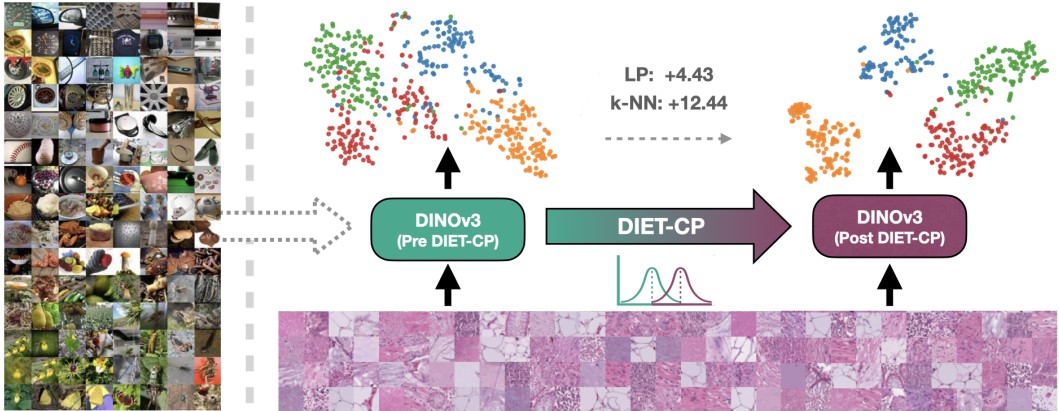

Figure 1: DIET-CP is a label-free and efficient method for steering foundation models towards a data distribution of interest, improving class separability in the embedding space and leading to improved unsupervised and linear probing performance. t-SNE plots are generated from a PathMNIST subset. Image credit: ImageNet [10] and PathMNIST [11]

performance boost. On medical image classification, we improve the F1 classification performance of DINOv2 and DINOv3 by 17.77 and 12.44 on k-NN, and 4.81 and 4.43 absolute percentage points on linear probing, from only a small amount of target data and no labels.

## 2 The DIET for Self Supervised Continued Pretraining

Our proposed method refines the representations of a foundation model in a self-supervised setting using cross entropy on the Datum IndEx as Target for Continual Pretraining (DIET-CP) [8]. The formulation of the continued pretraining loss for a backbone $f_{\boldsymbol{\theta}}$ is as follows:

$$\mathcal{L}_{\mathrm{DIET}}(\boldsymbol{x}_n) \;=\; \mathrm{XEnt}\big(\boldsymbol{W}\,f_{\boldsymbol{\theta}}(\boldsymbol{x}_n),\,n\big), \qquad \boldsymbol{x}_n \in \mathbb{R}^D, \tag{1}$$

where $n$ is the one-hot encoded index of each datum, meaning $n = 1$ for the first image, $n = D$ for the last image of a dataset of size $D$. $W$ represents a linear classification head for the DIET loss on the [CLS] token or mean-pooled patch representations.

This simple objective is an effective pretraining strategy for small datasets. Recent theoretical insights show that DIET's instance discrimination objective recovers ground truth factors of the underlying data generation process under certain assumptions, provably yielding linearly decodable representations [12]. For continual pretraining, DIET-CP offers the following benefits: (1.) no teacher checkpoints or other auxiliary parameters are need to continue pretraining, as the DIET loss requires no projector network or self-distillation. (2.) DIET-CP is effective with only a small number of training samples, and as little as 500-1000 samples can be sufficient for a considerable performance increase, as demonstrated in the experiments below. (3.) Compared to supervised finetuning, no additional hyperparameters are introduced. DIET-CP can be performed with the same parameters as any supervised finetuning strategy. This is especially crucial for the low-data regime we are investigating here, where few samples and even fewer labels are available and cross-validation of SSL hyperparameters may become intractable.

Two optimizations can be applied independent of dataset or backbone choice: DIET benefits from label smoothing on the cross-entropy loss [8], but contrary to training from scratch, we found that DIET-CP performs best with lower label smoothing values in our setup($\sim 0.3$). Further, to initialize $W$ without adversely affecting the backbone, DIET-CP can be started with a frozen backbone for the first steps.

Table 1: F1 classification performance on medical datasets before and after DIET continual pretraining using k-NN and linear probing, averaged over three runs.

| Backbone | Dataset | Pre DIET-CP (F1) | | Post DIET-CP (F1) | |
|---|---|---|---|---|---|
| | | k-NN | LP | k-NN | LP |
| DINOv2 | BreastMNIST | 64.89 | 82.21 | 88.54 (+23.66) | 88.90 (+6.69) |
| | DermaMNIST | 21.13 | 40.45 | 41.85 (+20.72) | 53.21 (+12.76) |
| | OCTMNIST | 41.57 | 71.05 | 74.89 (+33.32) | 85.41 (+14.37) |
| | OrganaMNIST | 57.17 | 78.51 | 72.37 (+15.20) | 80.30 (+1.79) |
| | OrgancMNIST | 58.30 | 76.49 | 72.40 (+14.10) | 79.02 (+2.53) |
| | OrgansMNIST | 46.74 | 62.47 | 57.46 (+10.72) | 62.21 (-0.26) |
| | PathMNIST | 84.15 | 93.17 | 94.53 (+10.38) | 95.94 (+2.77) |
| | PneumoniaMNIST | 63.67 | 89.29 | 93.43 (+29.75) | 95.93 (+6.64) |
| | RetinaMNIST | 39.91 | 50.05 | 41.95 (+2.04) | 46.06 (-3.99) |
| | Average | 53.06 | 71.52 | 70.82 (+17.77) | 76.33 (+4.81) |
| DINOv3 | BreastMNIST | 72.40 | 81.92 | 87.80 (+15.40) | 91.78 (+9.86) |
| | DermaMNIST | 22.50 | 47.26 | 33.92 (+11.42) | 50.52 (+3.26) |
| | OCTMNIST | 47.77 | 75.44 | 73.58 (+25.82) | 85.02 (+9.58) |
| | OrganaMNIST | 71.53 | 87.00 | 80.74 (+9.20) | 88.33 (+1.33) |
| | OrgancMNIST | 70.48 | 78.06 | 77.61 (+7.14) | 84.57 (+6.50) |
| | OrgansMNIST | 60.21 | 64.15 | 67.44 (+7.23) | 71.95 (+7.81) |
| | PathMNIST | 86.34 | 93.88 | 93.35 (+7.01) | 95.30 (+1.41) |
| | PneumoniaMNIST | 73.38 | 91.72 | 92.68 (+19.31) | 96.08 (+4.36) |
| | RetinaMNIST | 38.85 | 53.52 | 48.27 (+9.41) | 49.25 (-4.27) |
| | Average | 60.38 | 74.77 | 72.82 (+12.44) | 79.20 (+4.43) |
| MAE | BreastMNIST | 59.33 | 77.11 | 75.76 (+16.43) | 78.46 (+1.35) |
| | DermaMNIST | 22.90 | 33.23 | 30.43 (+7.52) | 39.87 (+6.64) |
| | OCTMNIST | 31.79 | 46.49 | 48.81 (+17.02) | 66.92 (+20.44) |
| | OrganaMNIST | 52.98 | 69.37 | 72.31 (+19.33) | 78.69 (+9.32) |
| | OrgancMNIST | 45.58 | 64.88 | 64.05 (+18.47) | 71.17 (+6.28) |
| | OrgansMNIST | 38.37 | 48.94 | 51.95 (+13.58) | 60.98 (+12.04) |
| | PathMNIST | 73.01 | 85.24 | 87.51 (+14.50) | 91.76 (+6.52) |
| | PneumoniaMNIST | 83.93 | 88.92 | 92.85 (+8.92) | 93.34 (+4.42) |
| | RetinaMNIST | 25.06 | 31.22 | 34.66 (+9.61) | 39.63 (+8.41) |
| | Average | 48.10 | 60.60 | 62.04 (+13.93) | 68.98 (+8.38) |

## 2.1 Experiments

The effect of using DIET continued pretraining is evaluated on a series of classification datasets that are both *in-domain* (natural images), and *out-of-domain* (medical images, optical astronomical images) for three pretrained vision foundation models.

We run Eq. (1) as continued pretraining on the fine-tuning dataset to align the foundation model to the target distribution. We start by training only $W$ for the first 5% of the epochs as described above. Afterwards, we unfreeze the last two blocks of the backbone and train them jointly with $W$ over a total of 150 epochs with 10% learning rate warmup and cosine annealing. More information and loss curves can be found in the appendix. Due to this simple setup, DIET-CP is very fast on a single GPU (<10 min. for ViT-B on an H100). For each task, we use DIET continued pretraining on a random subset of the training data ($N = 1000$, less for BreastMNIST and Galaxy10 DECals, see Appendix) and we record $k$-NN and linear probing metrics on the validation set before and after training on the subset. We report the F1 score due to class imbalance (see appendix for accuracy and dataset stats).

**Pretrained Backbones.** We evaluate the method on three popular pretrained vision encoders. DINOv2 [7] is a family of models trained via teacher–student self-distillation using a refined iBOT method [13]. DINOv3 [6] represents the latest version of this method, using a larger dataset and a further refined pretraining strategy to yield more robust and high-resolution features. Lastly, we use the popular masked-autoencoder (MAE) by He et al. [14] trained on ImageNet22k [10]. All models are ViT-B architectures [15] and initialized from publicly released checkpoints.

Table 2: Linear Probing and $k$-NN classification performance before and after DIET-CP (F1) for non-medical datasets. FGVC-Aircraft and Food-101 are considered *in-domain* fine-grained visual categorization tasks, while Galaxy10-DECaLS is an *out-of-domain* optical telescope imaging dataset.

| Backbone | Eval (F1) | FGVC-Aircraft | | Food-101 | | Galaxy10-DECaLS | |
|---|---|---|---|---|---|---|---|
| | | Pre | Post | Pre | Post | Pre | Post |
| DINOv2 | k-NN | 19.59 | 30.91 (+11.31) | 58.64 | 60.33 (+1.69) | 30.53 | 58.30 (+27.77) |
| | LP | 43.47 | 38.47 (-5.00) | 73.54 | 65.29 (-8.25) | 49.30 | 64.31 (+15.01) |
| DINOv3 | k-NN | 38.91 | 31.83 (-7.08) | 63.37 | 58.03 (-5.34) | 42.45 | 52.09 (+9.64) |
| | LP | 61.00 | 48.56 (-12.44) | 77.58 | 68.98 (-8.60) | 57.43 | 62.98 (+5.54) |
| MAE | k-NN | 3.74 | 6.83 (+3.09) | 3.73 | 11.92 (+8.19) | 20.44 | 33.93 (+13.49) |
| | LP | 6.77 | 11.54 (+4.77) | 10.41 | 21.10 (+10.69) | 26.98 | 38.94 (+11.96) |

Figure 2: Ablation study over the number of samples used for DIET-CP of a DINOv2 ViT-S. For training the $k$-NN and LP classifiers, a constant set of 1000 labels is used.

**Datasets.** As a highly relevant *out-of-domain* application, we cover a diverse set of medical imaging datasets, using a subset of MedMNISTv2 [16, 17]. The datasets vary in size and class imbalance and span various medical modalities: BreastMNIST (ultrasound, benign vs. malignant) [18], DermaMNIST (7-class dermoscopy) [19, 20], OCTMNIST and RetinaMNIST (retinal OCT and diabetic retinopathy grading) [21], OrganAMNIST/CMNIST/SMNIST (11-class organ recognition from CT in axial/coronal/sagittal views) [22, 23], PathMNIST (9-class colorectal histology) [11], and PneumoniaMNIST (binary pediatric chest X-ray) [24]. Further, we evaluate DIET-CP on Galaxy10 DECaLS, a 10-class optical telescope imaging dataset of galaxy morphologies [25, 26]. Lastly, we include two natural image datsets that are *in-domain* for the pretrained backbones, but require fine-grained visual categorization into around 100 classes (FGVC-Aircraft [27] and Food-101 [28]).

**DIET-CP Improves out-of-domain performance on medical images and galaxy morphology classification.** Table 1 presents pre- and post DIET-CP performance on MedicalMNIST datasets. On average across all tasks, DINOv2 and DINOv3 improve linear probing (LP) performance by 4.81 and 4.43 absolute percentage on F1 respectively, and dramatically on $k$-NN by 17.77 and 12.44, demonstrating the effectiveness of DIET-CP for unsupervised clustering in particular. MAE is a weaker baseline, in particular on linear and $k$-NN evaluation, but improves considerably by 13.93 on $k$-NN and 8.38 on LP. RetinaMNIST is the only dataset where LP performance degrades for both DINO models and represents an interesting outlier case as the only ordinal regression task, while $k$-NN performance reliably improves for all models.

Results on non-medical datasets are shown in Table 2. Here, we consider FGVC-Aircraft and Food-101 as fine-grained *in-domain* tasks for the vision models, which are trained exclusively, or with a significant bias, on natural images, while the astronomical images of Galaxy10-DECaLS are considered *out-of-domain*. DIET-CP does not improve fine-grained in-domain performance for the strong DINO models (DINOv2 improves only on k-NN). MAE performance is increased by DIET-CP but remains low. Representing a non-medical *out-of-domain* task, DIET-CP improves Galaxy10-DECaLS performance strongly across all models for both LP and $k$-NN evaluation.

An ablation over the number of training samples for DIET-CP is presented in Figure 2, using a DINOv2 ViT-S. We observe that 1000 samples are sufficient for a clear performance gain on linear probing, while $k$-NN improves earlier. More samples did not yield additional benefits for our setup.

# 3   Conclusions and Future Work

DIET-CP is a simple and sample efficient method for steering foundation models towards a target domain via continual pretraining on a small dataset, leading to measurable improvements on downstream tasks that are out-of-domain for the original backbone. A number of limitations remain as avenues for future work, such as the need for label-free prediction metrics on when DIET-CP helps performance, or deteriorates, as observed in some cases for fine-grained in-domain tasks, which could be coupled to determining how many layers of the backbone should be trained. For out-of-domain tasks however, we find that DIET-CP is a fast, viable and effective solution for improving state-of-the art foundation models.

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

# A DIET

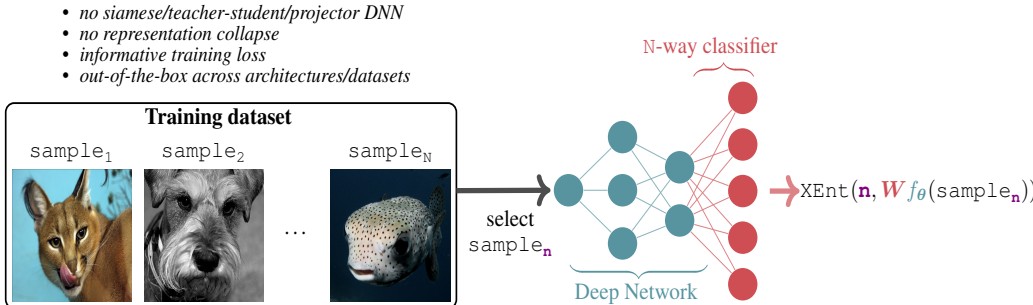

- *no siamese/teacher-student/projector DNN*
- *no representation collapse*
- *informative training loss*
- *out-of-the-box across architectures/datasets*

Figure 3: **DIET** uses the datum index (n) as the class-target –effectively turning unsupervised learning into a supervised learning problem. In our case, we employ the cross-entropy loss (X-Ent), no extra care needed to handle different dataset or architectures. As opposed to current SOTA, we do not rely on a projector nor positive views *i.e* no change needs to be done to any existing supervised pipeline to obtain DIET. Figure and caption from Ibrahim et al. [8], see original publication for more details.

# B Details on DIET-CP Setup

All experiments are performed using the same recipe. We use AdamW [29] over a total of 150 epochs with a 10% warmup to a learning rate of $1e-4$ and cosine annealing. For the first 5% of the epochs, the backbone remains frozen and only the DEIT head $W$ is trained. Afterwards, we unfreeze the last two transformer blocks and train them jointly with $W$. We use a batch size of 32 and a 0.05 weight decay. For each task, DIET continued pretraining is used on a random subset of the training data ($N = 1000$) and we record $k$-NN and linear probing metrics on the validation set before and after training on the subset.

All images are size 224x224 and are converted to RGB. We use positional embedding interpolation to adapt the ViTs to the input resolution.

The following augmentation pipeline is employed across all datasets:

```
v2.RGB
RandomResizedCrop(224, antialias=True),
RandomHorizontalFlip(),
RandomApply([transforms.ColorJitter(0.4, 0.4, 0.4, 0.2)], p=0.3)
RandomGrayscale(p=0.2),
RandomApply([transforms.GaussianBlur((3, 3), (1.0, 2.0))], p=0.2)
```

# C Additional Results

**DIET-CP loss versus performance.** Figure 4 presents DIET loss curves of a DINOv2 ViT-S plotted alongside k-NN and linear probing accuracy over three different MedMNIST tasks. The loss converges smoothly, but is not proportional to classification performance: DIET loss decreases monotonically even as linear probing and k-NN performance plateaus. A similar pattern is observed over different backbone types in Figure 5. These results highlight the need for for label-free metrics that better predict pretraining success.

**Additional classification results.** For the interested reader, Table 3 presents full k-NN and linear probing results as accuracy and F1 including standard deviation on MedMNIST tasks. We further show accuracy results for the non-medical datasets in Table 4 and note that Galaxy10_DECals is unbalanced in the class distribution.

**Ablation on backbone size.** A small ablation study on the backbone size is shown in Table 5, where we compare the performance of DINOv2 ViT-S versus ViT-B models on four datasets. Performance

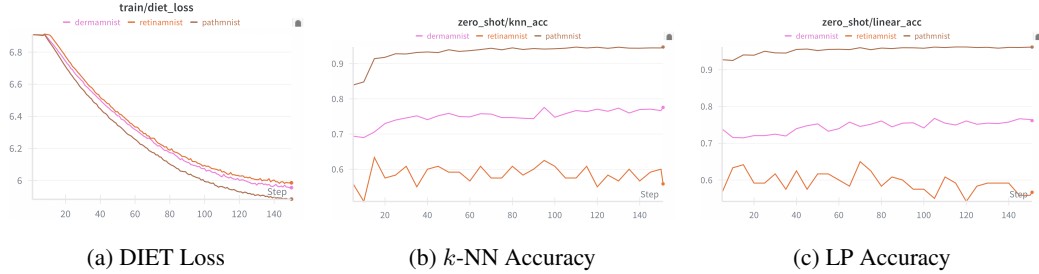

(a) DIET Loss     (b) $k$-NN Accuracy     (c) LP Accuracy

Figure 4: DIET loss curves for DINOv2 ViT-S and corresponding $k$-NN and linear probing accuracy on three MedMNIST datasets during training over 150 epochs.

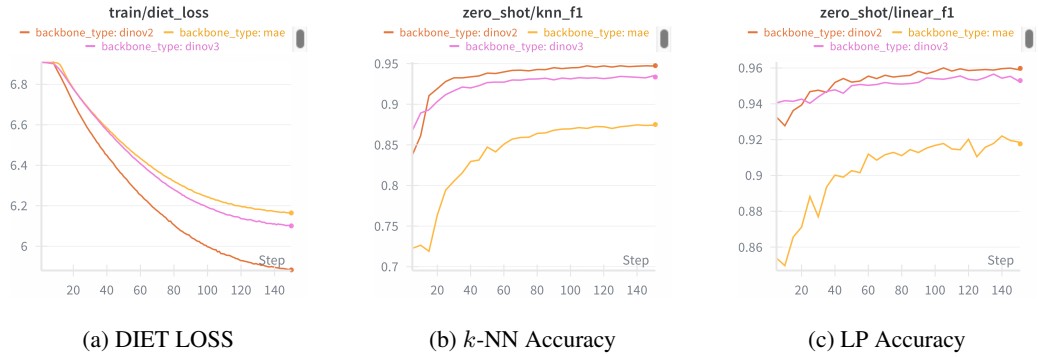

(a) DIET LOSS     (b) $k$-NN Accuracy     (c) LP Accuracy

Figure 5: DIET loss curves, $k$-NN and linear probing accuracy for ViT-B DINOv2, DINOv3, and MAE on PathMNIST. Backbones reach different loss levels, but they are not strongly correlated to downstream performance.

is measured as F1 score for k-NN and linear probing and averaged over three runs. As expected, the small model performs worse on average. Interestingly, the larger model also benefits more from DIET-CP, prompting further investigation into the scalability on larger models.

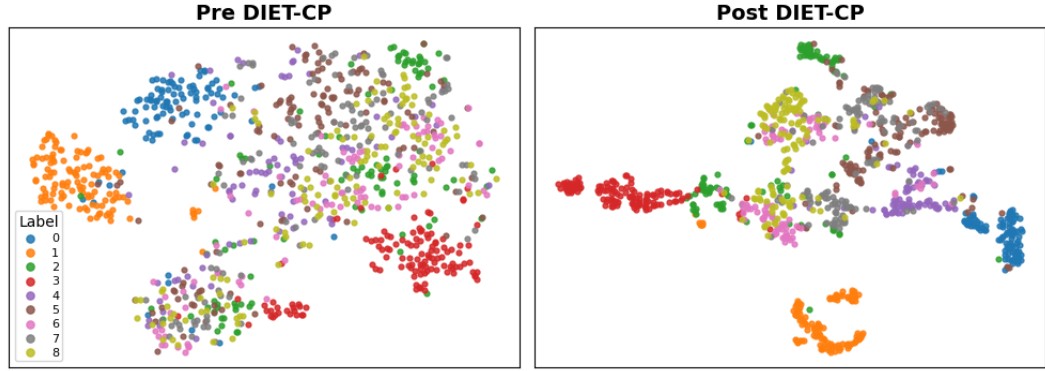

Figure 6: t-SNE plot of pre and post DIET-CP representations for MAE on PathMNIST.

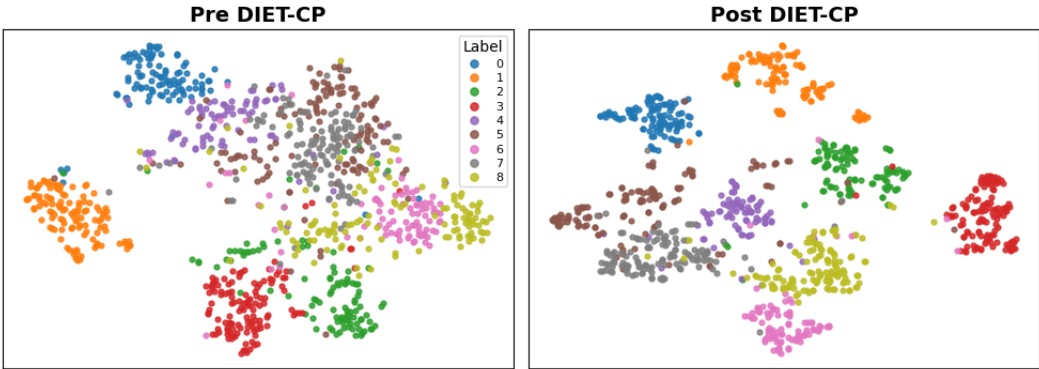

Figure 7: t-SNE plot of pre and post DIET-CP representations for DINOv2 on PathMNIST.

# D    Dataset Statistics

A dataset overview is provided in Table 6, including number of classes, images and class balance. Most of the datasets used in the analyses feature class imbalance. BreastMNIST contains less then 1000 images, the number of DIET classes is therefore equal to the training split ($N = 546$) for this data split. Similarly, as we train on a random 50% sample, we use $N = 800$ DIET-Classes for Galaxy10-DECaLS.

Table 3: Full results table for medical datasets with F1 and accuracy and standard deviation on $k$-NN and linear probe evaluation pre and post DIET-CP continued pretraining.

| Backbone | Dataset | Pre DIET-CP | | | | Post DIET-CP | | | |
|---|---|---|---|---|---|---|---|---|---|
| | | KNN Acc. | KNN F1 | Linear Acc. | Linear F1 | KNN Acc. | KNN F1 | Linear Acc. | Linear F1 |
| DINOv2 | breastmnist | 79.91±0.74 | 64.89±1.68 | 86.75±6.06 | 82.21±8.50 | 91.45±0.74 | 88.54±0.88 | 91.03±2.56 | 88.90±2.87 |
| | dermamnist | 68.99±0.42 | 21.13±2.79 | 71.98±0.28 | 40.45±1.73 | 77.87±0.42 | 41.85±1.63 | 76.02±0.35 | 53.21±0.80 |
| | octmnist | 73.73±0.79 | 41.57±0.56 | 84.67±0.10 | 71.05±1.44 | 87.99±0.22 | 74.89±0.12 | 92.08±0.13 | 85.41±0.67 |
| | organamnist | 63.74±3.30 | 57.17±2.07 | 80.91±2.00 | 78.51±2.19 | 77.93±1.95 | 72.37±3.55 | 81.03±1.62 | 80.30±1.52 |
| | organcmnist | 63.04±3.96 | 58.30±0.31 | 80.31±2.60 | 76.49±1.67 | 78.70±0.33 | 72.40±1.54 | 82.88±0.09 | 79.02±0.43 |
| | organsmnist | 54.16±3.29 | 46.74±4.18 | 67.90±1.79 | 62.47±1.38 | 63.03±0.84 | 57.46±2.24 | 66.25±1.59 | 62.21±1.22 |
| | pathmnist | 84.10±0.70 | 84.15±0.71 | 93.19±0.40 | 93.17±0.44 | 94.41±0.48 | 94.53±0.46 | 95.88±0.45 | 95.94±0.44 |
| | pneumoniamnist | 64.31±3.24 | 63.67±2.93 | 91.13±1.48 | 89.29±1.58 | 94.75±0.40 | 93.43±0.46 | 96.85±0.67 | 95.93±0.90 |
| | retinamnist | 58.75±6.48 | 39.91±1.44 | 61.67±3.54 | 50.05±3.90 | 57.08±1.77 | 41.95±6.35 | 57.92±2.95 | 46.06±3.46 |
| | Average | 67.86±2.55 | 53.06±1.85 | 79.83±2.03 | 71.52±2.54 | 80.36±0.79 | 70.82±1.91 | 82.22±1.16 | 76.33±1.37 |
| DINOv3 | breastmnist | 82.48±1.48 | 72.40±5.42 | 87.18±1.28 | 81.92±1.93 | 90.60±2.67 | 87.80±3.44 | 93.59±1.28 | 91.78±1.75 |
| | dermamnist | 70.56±0.42 | 22.50±1.24 | 73.65±1.36 | 47.26±2.33 | 74.78±1.04 | 33.92±1.69 | 77.40±0.72 | 50.52±1.90 |
| | octmnist | 76.36±0.11 | 47.77±0.54 | 85.78±2.38 | 75.44±3.25 | 87.47±0.67 | 73.58±2.85 | 91.66±0.42 | 85.02±0.05 |
| | organamnist | 75.49±3.21 | 71.53±4.35 | 87.13±1.04 | 87.00±1.46 | 84.83±1.76 | 80.74±2.56 | 89.30±1.41 | 88.33±1.11 |
| | organcmnist | 77.01±1.96 | 70.48±1.59 | 81.37±2.06 | 78.06±2.27 | 83.25±0.32 | 77.61±1.35 | 87.47±1.68 | 84.57±3.06 |
| | organsmnist | 65.31±0.32 | 60.21±0.46 | 68.27±1.44 | 64.15±0.30 | 72.72±0.35 | 67.44±0.39 | 76.24±0.09 | 71.95±0.63 |
| | pathmnist | 90.52±7.24 | 86.34±1.11 | 93.93±0.35 | 93.88±0.28 | 93.29±0.39 | 93.35±0.37 | 95.31±0.36 | 95.30±0.34 |
| | pneumoniamnist | 74.87±5.56 | 73.38±5.09 | 93.32±0.50 | 91.72±0.60 | 94.15±0.58 | 92.68±0.65 | 96.95±0.19 | 96.08±0.22 |
| | retinamnist | 57.78±2.93 | 38.85±4.35 | 63.61±2.10 | 53.52±1.78 | 60.28±1.73 | 48.27±1.30 | 58.61±1.27 | 49.25±2.56 |
| | Average | 74.49±2.58 | 60.38±2.68 | 81.58±1.39 | 74.77±1.58 | 82.37±1.06 | 72.82±1.62 | 85.17±0.82 | 79.20±1.29 |
| MAE | breastmnist | 76.07±0.74 | 59.33±0.75 | 84.62±1.28 | 77.11±1.53 | 82.48±1.96 | 75.76±2.99 | 83.76±0.74 | 78.46±1.18 |
| | dermamnist | 69.92±0.55 | 22.90±1.32 | 72.08±1.41 | 33.23±4.00 | 73.45±0.21 | 30.43±2.06 | 74.01±1.12 | 39.87±3.31 |
| | octmnist | 60.42±1.98 | 31.79±1.79 | 73.22±0.59 | 46.49±2.66 | 77.89±0.79 | 48.81±1.40 | 82.19±0.99 | 66.92±1.01 |
| | organamnist | 62.97±4.22 | 52.98±2.18 | 73.32±0.45 | 69.37±1.38 | 76.56±0.82 | 72.31±1.19 | 80.56±2.79 | 78.69±2.34 |
| | organcmnist | 54.29±2.61 | 45.58±3.11 | 69.72±3.09 | 64.88±4.19 | 70.74±2.29 | 64.05±1.82 | 77.01±1.17 | 71.17±0.98 |
| | organsmnist | 47.94±3.32 | 38.37±5.18 | 56.00±4.90 | 48.94±7.13 | 58.14±2.05 | 51.95±1.94 | 67.17±0.23 | 60.98±0.08 |
| | pathmnist | 73.96±1.72 | 73.01±1.20 | 85.41±0.49 | 85.24±0.75 | 87.53±0.64 | 87.51±0.62 | 91.78±0.23 | 91.76±0.31 |
| | pneumoniamnist | 86.07±1.08 | 83.93±1.24 | 90.94±0.40 | 88.92±0.60 | 94.37±0.13 | 92.85±0.14 | 94.75±0.13 | 93.34±0.18 |
| | retinamnist | 47.92±0.59 | 25.06±2.87 | 50.42±0.59 | 31.22±1.22 | 53.33±0.00 | 34.66±0.60 | 55.00±0.00 | 39.63±1.96 |
| | Average | 64.39±1.87 | 48.10±2.18 | 72.86±1.47 | 60.60±2.61 | 74.94±0.99 | 62.04±1.42 | 78.47±0.82 | 68.98±1.26 |

Table 4: Accuracy comparison before and after DIET-CP for non-medical datasets. Improvements (in parentheses) are green for positive, red for negative, and gray if $|\Delta| < 1.0$.

| Backbone | Dataset | Pre DIET-CP (Acc.) | | Post DIET-CP (Acc.) | |
|---|---|---|---|---|---|
| | | k-NN | LP | k-NN | LP |
| dinov2 | fgvc_aircraft | 21.81 | 44.74 | 32.52 (+10.71) | 39.48 (-5.26) |
| | food101 | 61.59 | 74.02 | 61.79 (+0.20) | 65.82 (-8.21) |
| | galaxy10_decals | 37.16 | 54.07 | 64.57 (+27.40) | 67.64 (+13.57) |
| dinov3 | fgvc_aircraft | 42.85 | 62.18 | 34.42 (-8.43) | 49.47 (-12.70) |
| | food101 | 65.91 | 77.89 | 60.25 (-5.65) | 69.38 (-8.51) |
| | galaxy10_decals | 49.65 | 62.05 | 59.60 (+9.95) | 66.67 (+4.62) |
| mae | fgvc_aircraft | 4.60 | 7.41 | 7.87 (+3.27) | 11.92 (+4.51) |
| | food101 | 4.20 | 11.00 | 13.20 (+9.00) | 21.46 (+10.45) |
| | galaxy10_decals | 24.52 | 33.12 | 40.46 (+15.95) | 43.27 (+10.15) |

| Model Size | Dataset | Pre DIET-CP (F1) | | Post DIET-CP (F1) | |
|---|---|---|---|---|---|
| | | k-NN | LP | k-NN | LP |
| Small | BreastMNIST | 77.27±2.18 | 84.48±0.71 | 83.01±4.12 (+5.74) | 87.58±0.75 (+3.10) |
| | DermaMNIST | 23.68±1.09 | 43.11±4.13 | 31.83±1.78 (+8.15) | 44.44±1.27 (+1.32) |
| | FGVC-Aircraft | 19.49±0.61 | 39.80±1.04 | 27.68±1.17 (+8.20) | 35.76±0.45 (-4.04) |
| | OctMNIST | 44.92±0.92 | 73.00±0.84 | 71.65±2.40 (+26.73) | 81.45±0.55 (+8.46) |
| | OrganAMNIST | 65.32±4.83 | 79.53±2.85 | 79.07±3.44 (+13.75) | 83.89±2.37 (+4.36) |
| | **Average** | 46.14 | 63.98 | 58.65 (+12.51) | 66.62 (+2.64) |
| Base | BreastMNIST | 64.89±1.68 | 82.21±8.50 | 88.54±0.88 (+23.66) | 88.90±2.87 (+6.69) |
| | DermaMNIST | 21.79±2.27 | 40.86±1.41 | 41.47±1.33 (+19.68) | 53.02±0.65 (+12.17) |
| | FGVC-Aircraft | 19.59±0.09 | 43.47±0.16 | 30.91±1.60 (+11.31) | 38.47±0.78 (-5.00) |
| | OctMNIST | 41.57±0.56 | 71.05±1.44 | 74.89±0.12 (+33.32) | 85.41±0.67 (+14.37) |
| | OrganAMNIST | 57.17±2.07 | 78.51±2.19 | 72.37±3.55 (+15.20) | 80.30±1.52 (+1.79) |
| | **Average** | 41.00 | 63.22 | 61.63 (+20.63) | 69.22 (+6.00) |

Table 5: DINOv2 model size ablation: Performance comparison before and after DIET-CP across small and base model variants. Improvements shown in parentheses.

Table 6: Information on the number of samples and classes in the datasets used for experiments. All datasets, except for Food-101 and FGVC-Aircraft are unbalanced. If no official validation split is defined, we sample a random 50% split from the training set.

| Dataset | Classes | Train | Val | Test | Class balance |
|---|---|---|---|---|---|
| FGVC-Aircraft | 102 | 3400 | 3400 | 3400 | balanced |
| Food-101 | 101 | 75750 | - | 25250 | balanced |
| Galaxy10-DECaLS | 10 | 1600 | - | 1736 | skewed |
| BreastMNIST | 2 | 546 | 78 | 156 | skewed |
| DermaMNIST | 7 | 7007 | 1003 | 2005 | skewed |
| OCTMNIST | 4 | 97477 | 10832 | 1000 | skewed |
| RetinaMNIST | 5 | 1080 | 120 | 400 | skewed |
| OrganAMNIST (axial) | 11 | 34561 | 6491 | 17778 | skewed |
| OrganCMNIST (coronal) | 11 | 12975 | 2392 | 8216 | skewed |
| OrganSMNIST (sagittal) | 11 | 13932 | 2452 | 8827 | skewed |
| PathMNIST | 9 | 89996 | 10004 | 7180 | skewed |
| PneumoniaMNIST | 2 | 4708 | 524 | 624 | skewed |

