# OpenReview forum: "DIET-CP: Lightweight and Data Efficient Self Supervised Continued Pretraining"
_NeurIPS.cc/2025/Workshop/UniReps — UniReps2025_

### Official Review · Reviewer_5oeD · 2025-09-12

**Confidence:** 3

**Review:**

In this paper, the authors proposed a simple but effective way for continued pre-training: attach a linear head on top of the pre-trained model and assign each training data a one-hot label, and then use cross entropy loss to train the model to predict the one-hot label of each input training data. The proposed method is label-free and very simple and cheap to train. The proposed method is evaluated on a subset of MedMNIST as well as 3 other datasets, and evaluation results show that the proposed method is able to improve domain-specific performance in most of the evaluated tasks.

Strengths:

1. The proposed method is very simple and easy to implement. It does not require labels or teacher-student framework, and runs fairly quickly.
2. The proposed method yields significant domain-specific performance gains despite being simple.
3. The method is tested on many different domains and on 3 different pre-trained models, demonstrating a certain level of generalizability.
4. The paper explored the correlation between performance and number of samples used for DIET continued pre-training.

Weaknesses:

1. In all experiments, the method is only compared to no continued pre-training. It is unclear how the method's performance compares to any other continued pre-training methods or direct self-supervision on the domain-specific training set. It's okay for the proposed method to underperform compared to other methods trained on the same domain-specific training set (since the proposed method is much simpler and cheaper), but the comparison is important for future researchers to better understand what is the performance/cost tradeoffs.
2. Another important aspect of the proposed method that needs further exploration is how performance scales with respect to the convergence of DIET objective, i.e. does the pre-trained model perform better with lower diet training loss? Or does it peak at a certain point and then overfits? In all experiments in the paper, the number of epochs of applying DIET is fixed to 150, which is not sufficient to fully explore the relationship between the loss and the performance (as shown in the screenshots in the appendix).

**Score:**

3

**Topic Fit:**

1

---

### Official Review · Reviewer_Mtts · 2025-09-14

**Confidence:** 4

**Review:**

**Summary**

The paper highlights the advantages of using the DIET self-supervised loss as a strategy for continual pre-training. The DIET loss treats all instances of the dataset as a single class and applies cross-entropy loss, considering the feature embedding \(X_{i}\) and its corresponding datum index \(y_{i} = i\). The evaluation includes three self-supervised pre-trained backbones (DINO-V2, DINO-V3, and MAE), with the target task being out-of-domain for the pre-trained backbone. The authors consider nine tasks from the medMNIST dataset, galaxy morphology classification, and out-of-distribution domains, as well as three standard computer vision tasks.

The evaluation compares KNN and linear probing (LP) performance before and after applying the proposed DIET-CP. The results suggest that their method improves both KNN and LP performance, especially when 500 to 1,000 unlabeled samples are available. However, the results are inconsistent when scaling the number of available samples, as demonstrated in Figure 2.


**Strengths**

The paper effectively demonstrates the benefits of applying the DIET self-supervised loss as a strategy for continual pre-training in out-of-domain target applications, considering the pre-trained backbone in both KNN and linear probing classification. Additionally, the paper is straightforward and easy to understand.

The ablations consider scenarios with different numbers of samples, up to 10,000 (Figure 2). The approach is simple, requires no modifications to the architecture, and can be easily integrated into many existing models and classification tasks. A significant advantage is that the results indicate improved performance in low-data regimes, particularly with up to 1,000 samples, which is beneficial for domain-specific cases, such as medical applications, where obtaining a high number of samples is challenging. Furthermore, training with only 1,000 samples is efficient and typically does not require much time.


**Weakness**

Although the paper presents promising results for the proposed approach, it does not compare its performance with other self-supervised losses, such as contrastive learning, feature decorrelation loss, or Focal Loss. At this point, it is unclear whether the performance gains arise from training on out-of-domain data or from the current loss. For instance, I would expect greater performance improvements in out-of-domain datasets, such as medical ones, because the features from pre-training on ImageNet are not fully reused [1].


**Suggestions**
- I suggest reformatting Table 1 to allow for easier comparison between the performances before and after DIET-CP, similar to Table 2. One idea is to create one column group containing two subgroups.
- It may be possible to maintain in-distribution performance without degrading it by fine-tuning only a subset of layers rather than all or a large number of layers.
- I highly recommend that the authors perform ablation studies on other self-supervised approaches or feature-only losses, such as [2, 3]. Additionally, works of [4, 5] can also provide extra insights about why results are mostly positive for out-of-distribution data and imbalanced datasets.
- Since the authors sample up to 1,000 from the target task, did they analyze how this subset affects final performance? I also suggest reporting the standard deviation across multiple runs and varying the samples within the training data.

**References**

[1] Matsoukas, Christos, et al. "What makes transfer learning work for medical images: Feature reuse & other factors.", Conference on Computer Vision and Pattern Recognition, 2022.

[2] Madaan, Divyam, et al. "Representational Continuity for Unsupervised Continual Learning.", International Conference on Learning Representations, 2022

[3] Bardes, Adrien, Jean Ponce, and Yann LeCun. "VICReg: Variance-Invariance-Covariance Regularization for Self-Supervised Learning.", International Conference on Learning Representations, 2022.

[4] Konstantakos, Sotirios, et al. "Self-supervised visual learning in the low-data regime: a comparative evaluation.", Neurocomputing, 2025.

[5] Liu, Hong, et al. "Self-supervised Learning is More Robust to Dataset Imbalance.", International Conference on Learning Representations, 2022*.

**Score:**

2

**Topic Fit:**

2

---

### Official Review · Reviewer_aSh9 · 2025-09-15
**DIET-CP: Lightweight and Data Efficient Self Supervised Continued Pretraining.**

**Confidence:** 4

**Review:**

The paper is very clearly written, with precise and efficient explanations. The paper introduces DIET-CP, a lightweight self-supervised strategy for continued pretraining of vision foundation models using only a small set of unlabeled target-domain data. By applying a simple cross-entropy loss over instance indices, the method avoids complex components like teacher models, contrastive pairs, or projector heads. It proves especially effective in domains like medical imaging and astronomy, where data is scarce and labels are expensive. The method is validated across multiple pretrained backbones (DINOv2, DINOv3, MAE) and datasets, showing consistent improvements in k-NN and linear probing (LP) performance, using only 500–1000 samples and <10 minutes of compute on a single GPU.

Strengths:
1. The method is simple and practical; it requires only pretrained backbone weights and does not depend on labels, teacher models, or optimizer state. It uses a simple and well-established instance-level loss based on cross-entropy over sample indices.

2. The method yields substantial improvements, with up to +33.32 F1 in k-NN classification and +20.44 F1 in linear probing, particularly on out-of-domain tasks such as medical and astronomical imaging.

3. The authors evaluate DIET-CP across a diverse set of datasets, including 9 medical, 3 out-of-domain, and 2 in-domain fine-grained benchmarks, using 3 different pretrained backbones (DINOv2, DINOv3, MAE).

4. The full training process completes in under 10 minutes on a single GPU for ViT-B models, and the method incurs no additional memory or architectural overhead compared to supervised fine-tuning.

Weakness:
1. DIET-CP is essentially an instance discrimination loss with cross-entropy over image indices, which is not new in itself (e.g., InstDisc, InfoNCE). The novelty lies more in the use case and application to continued pretraining, not in the algorithmic core.

2. On certain in-domain fine-grained classification tasks like FGVC-Aircraft and Food-101, the method results in reduced performance, with LP F1 drops of up to -12.44, and the paper does not offer guidance on when DIET-CP is likely to help or harm.

3. It would strengthen the case to compare with other lightweight domain adaptation or continued pretraining methods like LoRA / Adapter tuning, contrastive learning baselines (SimCLR, MoCo) or linear probe with frozen features only.


Significance of this work is High.
This work provides a realistic path to unlock the potential of foundation models in low-resource domains, without labels and without full pretraining complexity. It provides a scalable, practical, and high-performing solution in domains like medical imaging or astronomy, where labeled data is scarce, and compute is limited

**Score:**

4

**Topic Fit:**

3

---

### Official Review · Reviewer_4GS4 · 2025-09-16
**Simple, low-data CPT with instance discrimination: strong k-NN gains, mixed LP on fine-grained**

**Confidence:** 4

**Review:**

The paper proposes DIET-CP, a CPT recipe that adds an instance-discrimination CE head on top of a frozen vision backbone, then unfreezes the last two blocks to co-train on a small unlabeled subset (typically N=1000). It aims to be teacher-free, projector-free, and hyperparameter-free. Experiments on MedMNIST, Galaxy10, and two fine-grained natural-image datasets show large k-NN F1 gains and smaller/mixed linear-probe changes; regressions appear on in-domain Food-101/FGVC and RetinaMNIST LP.

**Pros**:
1. Works with just backbone weights; no teacher/projector; fast (<10 min for ViT-B on H100). Good for use-cases with tiny target datasets.
2. Strong k-NN improvements across medical/astronomy; method matches its objective (improving local neighborhood structure).
3. Simple, replicable recipe with explicit optimizer/epochs/augmentations; useful starting point for the community.

**Cons**:
1. Missing side-by-side with other cheap CPT baselines (e.g., SimCLR-like) and with few-label supervised CPT (finetuning last blocks with 100–1000 labels). Can’t tell from the experiments if DIET-CP is the best cheap choice or just a cheap choice.
2. Scope of "any strong model" claim feels broad, given that there are regressions on in-domain datasets, which shows that the experiments didn't quantify catastrophic forgetting.
3. Building on the above point, no explicit check of performance on the original in-domain distribution post-CPT.
4. Theoretical pointer (linearly decodable factors from DIET) is cited but assumptions aren’t connected to the low-data CPT regime; it’s unclear when the guarantees meaningfully apply here.

**Suggestions for authors**:
1. Compare to an equally cheap (e.g. SimCLR) CPT (same data budget, same freeze/unfreeze)
2. Show forgetting vs. gain curves. Example experiments, CPT with DIET on 1k images and then evaluate on ImageNet-style validation.
3. Can you add label-free early-stopping or "should-I-CPT?" indicators (e.g., cluster metrics) to avoid the in-domain regressions you report?

**Overall**:
A useful, minimal CPT recipe that reliably improves unsupervised structure in small, out-of-domain settings. The idea is incremental but practical; the empirical story is honest about where it hurts.

**Score:**

3

**Topic Fit:**

3